# Renovation Potential Evaluation and Type Identification of Rural Idle Residential Land: A Case Study of Yuzhong County, Longzhong Loess Hilly Region, China

Libang Ma [1,2,3,*] , Tianmin Tao [1], Yao Yao [1] and Yawei Li [1]

1   College of Geography and Environmental Science, Northwest Normal University, Lanzhou 730070, China
2   Key Laboratory of Resource Environment and Sustainable Development of Oasis, Lanzhou 730070, China
3   Northwest Institute of Urban-Rural Development and Collaborative Governance, Lanzhou 730070, China
*   Correspondence: malb0613@nwnu.edu.cn

**Abstract:** The land problem is the key to the implementation of the rural revitalization strategy, and the land suitability evaluation is the basis for the renovation and classification of idle rural residential areas. Taking Yuzhong County in the Loess Hilly region of Longzhong, China as the research area, this study constructed an idle residential identification matrix by combining the evaluation results of ecological protection suitability, agricultural production suitability, and construction and development suitability, and realizing renovation zoning of rural residents through the advantage type identification method. The results show that: (1) The waste in rural residential is serious, and there are significant phenomena of "one household with two houses" and "one household with multiple houses." The renovation potential of 1700 idle rural residential patches is 1.18 km$^2$. (2) The spatial differences in the suitability of rural residents in Yuzhong County are significant, and through the rational guidance and planning layout of rural residential renovation, it can provide an important decision-making basis for the rational utilization of rural residents and national land–space planning. (3) The renovation of rural residential should be guided by national land–space planning; make a solid plan for the renovation of rural residential areas; formulate a scientific plan for the renovation, relocation, and reuse of rural "hollow houses"; stimulate the vitality of rural land resources; and promote the revitalization of the countryside and the improvement of the rural living environment.

**Keywords:** residential land right confirmation; idle residential land identification; renovation types; suitability; Yuzhong County; China

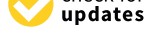



## 1. Introduction

The land issue is the key to the implementation of the rural revitalization strategy. Rural residential renovation is the key to land renovation planning, and the weak link and implementation difficulty of land renovation. The role of residential renovation has expanded from focusing on improving land-use efficiency to improve agricultural production conditions, employment, infrastructure, and public facilities, and promoting targeted poverty alleviation and rural revitalization [1–4]. Rural residential renovation has become an important way of realizing the intensive development of rural construction land, but the phenomenon of "de-ruralization" and "de-farmerization" in the implementation process has led to a certain gap between the goal and effect of rural residential renovation [5–7]. Therefore, it is necessary to actively explore the suitability evaluation of rural land and reasonable renovation zoning planning based on local practice and summarize the multiple paths to achieve the multidimensional objectives of the rural residential land, such as intensive and efficient rural residential land, integration and reorganization, market-oriented allocation. This can promote the revitalization of rural areas and the integrated development of urban and rural areas. From the perspective of land resource utilization and management, existing studies have studied the evaluation of rural idle residential land,

idle factors, land remediation, exit intention, and influencing factors, land use transformation, etc. [8–10]. There are also studies on rural residential land transfer, withdrawal, replacement, and the property rights system from the perspective of the economy and law [11–13]. This has accumulated a large number of research results for rural residential renovation and rural revitalization in China.

Renovation potential assessment and type identification are important aspects of the rural idle residential land management process. With the rapid advancement of urbanization, the social and economic structure in rural China has changed significantly. The urbanization process should be a process of transferring the rural population and reducing rural construction land and increasing of urban population and urban construction land. However, due to the rapid growth of the rural economy and lack of residential land exit mechanism, rural idle residential land has become a common phenomenon in rural areas in the process of rapid urbanization, the problem of idle residential land has aroused widespread concern from all walks of society, and idle residential land renovation potential has increased dramatically [14–17]. In the renovation potential assessment and type identification of rural idle residential land, academic circles have carried out a lot of research on the spatial and temporal characteristics of rural idle residential land expansion [18], the standard and measurement of idle residential land [19–21], the zoning and realization path of renovation potential [22,23], the renovation potential scale and composition characteristics [24], the human–land coupling in green transformation [25]. Local governments have also actively explored the path to promote the comprehensive renovation of villages, forming a model of "exchanging residential land with houses" in Tianjin City, "two separations and two exchanges" in Jiaxing City, and land tickets in Chongqing City [26]; in addition, Donggang City in Liaoning Province, Ganzhou City in Jiangxi Province [27], Yueyang City in Hunan Province, and other cities have carried out rural "hollow house" renovation practices [28]. These research and practices provide a reference for effective village renovation and the orderly withdrawal of residential land [28,29].

In conclusion, Scholars have not only studied the causes and exit mechanisms of idle residential land from the macro-scale, but also analyzed the farmers' willingness to organize their residential land [30] and influencing factors of residential consolidation from the micro-scale [31,32]. In terms of research content, it focuses more on the potential, influencing factors, policies, and experience and practices before and after renovation [27], while there are few studies on the renovation direction of idle residential based on the suitability evaluation of rural land. The suitability evaluation of rural residential land is the basis for the renovation and zoning of rural residential areas. The suitability evaluation of rural residential land comprehensively evaluates the regional matching degree and confirms its suitability according to the current land use situation and socioeconomic conditions of regional rural residential land, and the requirements of the new rural construction plan for the structure and space of rural residential land. Based on this, this paper took Yuzhong County in the Loess Hilly and Gully Region of Central Gansu in China as the research area, and based on the rural residential ownership confirmation data, the land use survey data, high-resolution remote sensing images, and on-the-spot investigation, constructed the identification matrix of the idle residential land, identified the spatial location and area of the idle residential land in 268 administrative villages of the whole county. In addition, based on the evaluation of the ecological protection suitability, agricultural production suitability, and construction and development suitability and in conjunction with the identification method of dominant types, the types of rural idle residential renovation were classified, and their renovation potential was calculated, which provides a scientific basis for the rational utilization of the rural idle residential land and the rural revitalization strategy.

## 2. Materials and Methods

### 2.1. Study Area

Yuzhong County is located from 103°49′15″ E to 104°34′40″ E and from 35°34′20″ N to 36°26′30″ N, which is the intersection of the Loess Plateau and Qinghai–Tibet Plateau

and three natural areas (Figure 1), with a length of 92 km from north to south and a width of 54 km from east to west, with a total area of 3302 km$^2$. Yuzhong County belongs to a temperate semi-arid climate, which is arid in spring, hot in summer, rainy and cool in autumn, and cold and dry in winter, with an average annual temperature of 7.5 °C, annual rainfall of 428 mm, evaporation of 1343.1 mm, and an average annual wind speed of 1.5 m/s. There are rich water resources, the annual total runoff of surface water is 53.49 × 10$^6$ m$^3$ and the total reserve of groundwater is 940 × 10$^6$ m$^3$. The altitude is 1480–3670 m, the terrain is high in the south and low in the north, and the middle is low-lying, showing a saddle shape.

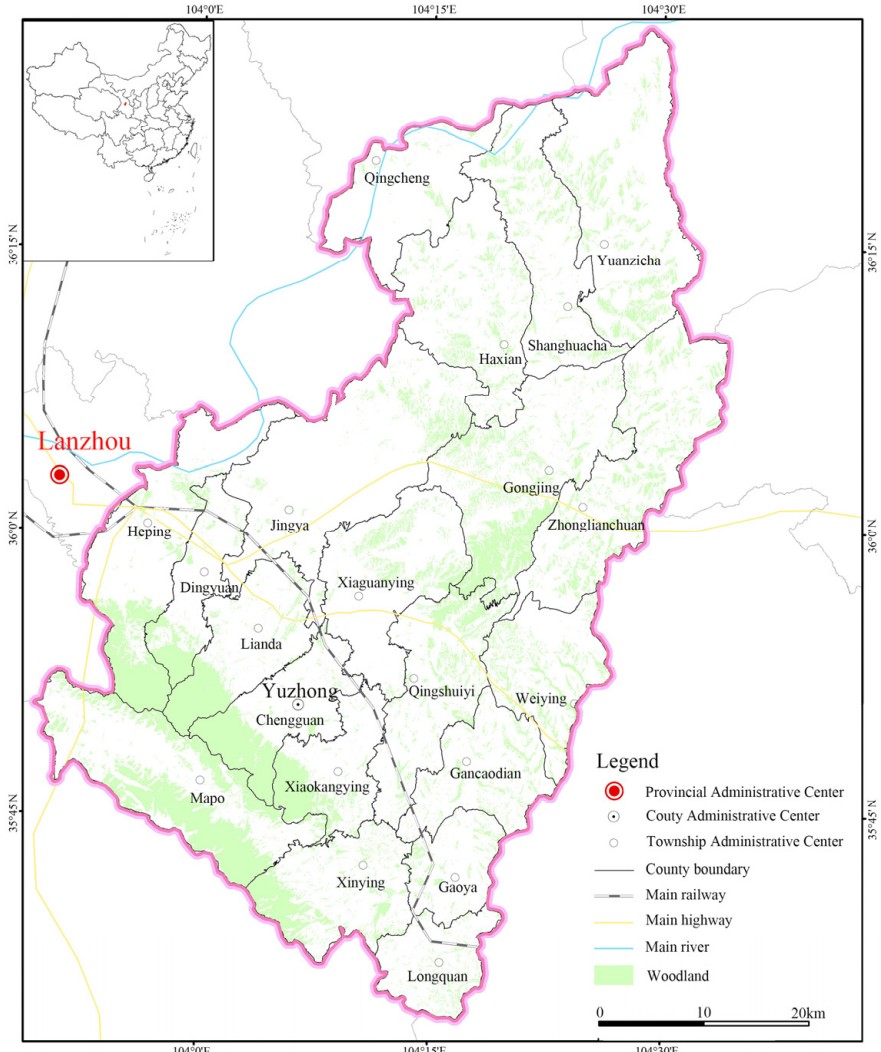

**Figure 1.** Location of Yuzhong County.

At the end of 2019, Yuzhong County had jurisdiction over 20 townships with a registered population of 461.10 × 10$^3$, among which the rural population was 351.90 × 10$^3$, accounting for 76.31% of the total registered population. The annual GDP was USD 2.23 × 10$^9$, and the regional fiscal revenue was USD 194.43 × 10$^6$. The first, second, and third industries have realized the added value of USD 201.88 × 10$^6$, USD 1.04 × 10$^9$, USD and 991.06 × 10$^6$, respectively. The per capita disposable income of urban residents is 2.24 times that of rural residents, with a large income gap between urban and rural areas. Yuzhong County is affected by the siphon effect of the provincial capital city, and the rural population mainly flows to Lanzhou City and Yuzhong County. Transitional geographical differences and differences in socioeconomic conditions have caused a large number of

rural populations to flow out, and the phenomenon of idle rural residential land in rural areas has become more and more popular.

*2.2. Data Sources and Research Framework*

2.2.1. Data Sources

The data in this study mainly come from the following five aspects:

(1) Vector data: the vector administrative boundary of Yuzhong County (1:250,000) originated from the Gansu Provincial Bureau of Surveying and Mapping; the road data came from the geospatial data cloud http://www.gscloud.cn/ (accessed on 30 June 2022); (2) natural geographic data: DEM in Yuzhong County, vegetation net primary productivity (NPP), and vegetation coverage (NDVI) data came from the geospatial data cloud. After the DEM was calibrated by ArcGIS, the elevation, surface relief, and slope gradient were extracted; (3) meteorological data: the daily precipitation and temperature data of Yuzhong County came from the Gansu Provincial Bureau of Meteorology and the average annual precipitation and temperature data of different regions were obtained by interpolation analysis of the meteorological station data using ArcGIS software; (4) land use data: the data of cultivated land quality grade, land use types, and soil texture attribute in Yuzhong County mainly came from the land use data of rural residential areas in Yuzhong County (the third land use survey data); the rural residential ownership confirmation data in 2019 came from the Natural Resources Bureau in Yuzhong County; (5) socio-economic data: the socioeconomic data of Yuzhong County, such as population, urbanization rate, and GDP, came from the *Yuzhong County Statistical Yearbook* and *Yuzhong County Government Work Report* (2020) [33].

Because the acquired data has the characteristics of being multi-precision, multi-source and multi-scale, the mathematical basis and precision of the same elements obtained from different data sources and different data acquisition methods are different [34]; therefore, we used ArcGIS to unify the coordinates, projection transformation, geometric correction and standardization of the data, and unified all the raster data to a resolution of $30 \times 30$ m, and carried out related researches on this basis.

2.2.2. Research Framework

From the perspective of land use characteristics, idle residential land refers to the idle situations caused by uninhabited or unused buildings or appendages on its ground in a certain period of time, mainly including the rural residential land of "One Household with Multiple Houses", building new houses without demolishing old ones, and houses with dilapidated appearance and collapsed houses [35,36]. In China, rural residential land refers to the collective construction land that rural villagers enjoy based on their membership in the collective economic organization (administrative village or production team) and that can be used to build houses. It implements the management system of "One Household with One Houses". Confirmation of the right to use residential land refers to the right of rural residents to occupy and use their own collective land for the construction of their own residential houses. "One Household with One House" should be insisted upon. The residential land occupied other than "One Household with One House" should not be confirmed for registration, and the occupied residential land exceeding the "One Household with One House" policy is regarded as illegally occupied land. In the investigation, it was found that most of the residential land that has not been confirmed for registration in Yuzhong County is "One Household with Two Houses", or even "One Household with Multiple Houses" in which the old houses have not been withdrawn and new ones have been built. This study assumes that they are idle residential land that can be renovated. This paper mainly includes three sections of research contents: identification of idle residential land, suitability evaluation of rural land, and classification of idle residential renovation types (Figure 2).

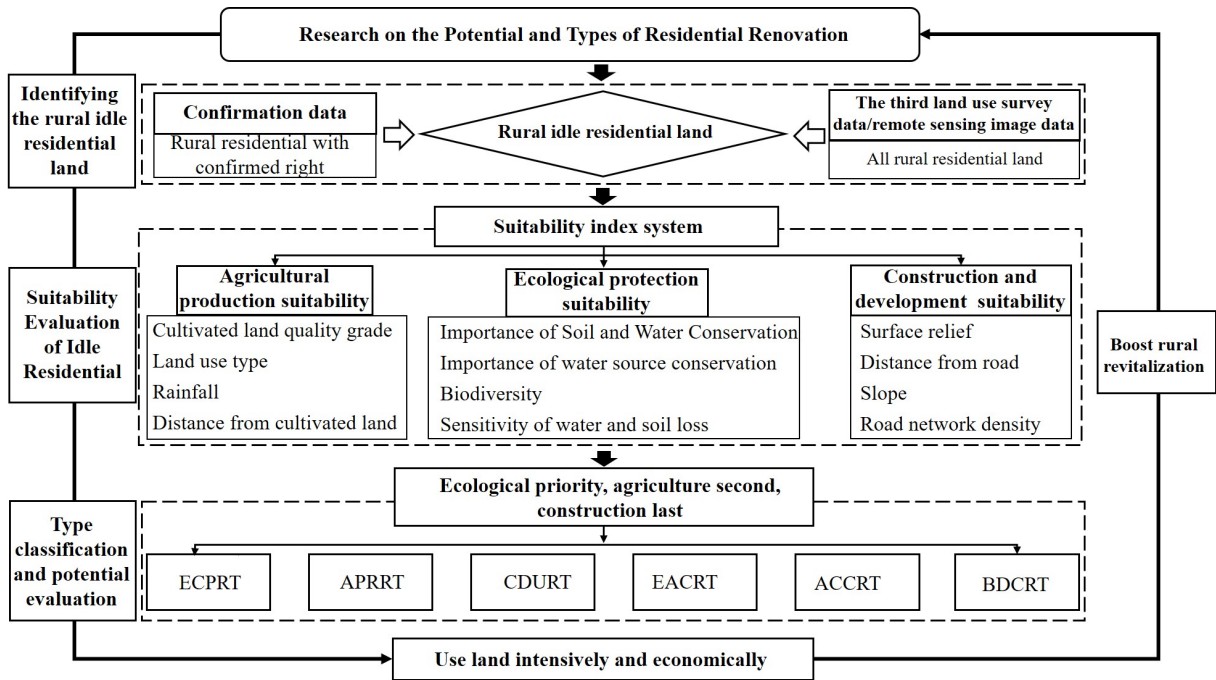

**Figure 2.** Research framework of residential renovation potential and types.

(1) Identification of idle residential land: In the survey, it was found that the rural population in Yuzhong County had a significant phenomenon of seasonally going out as migrant workers, resulting in a large number of seasonal vacant houses. However, rural residents have different time limits for migrant workers, so it is impossible to accurately judge whether the residential land is idle all year round, and there may be a phenomenon of misjudging the temporary idle houses as idle residential land. Confirmation of the right to use rural residential land can clarify the right to use rural residential land and deny the illegal and unreasonable residential land of the "One Household with Two Houses" and "One Household with Multiple Houses" and the like to be confirmed. Therefore, we took the confirmed residential land as the discriminant standard and combined the third land use survey data and the remote sensing image data of Yuzhong County to identify the idle residential land.

(2) Suitability evaluation of the rural land: Factors such as location conditions, resource endowment, environmental capacity, industrial base, and historical and cultural heritage should be taken into account in the utilization of the idle residential land, and the utilization mode of rural idle residential land suitable for local reality should be selected. This paper evaluates the suitability of the rural land from three aspects: ecological protection, agricultural production, and construction and development, and analyzes the spatial differentiation characteristics of each suitability of the idle residential land.

(3) Classification of the idle residential renovation types: based on the suitability evaluation of the idle residential renovation and taking the ecological suitability value as the first criterion of the classification, the classification was performed following the principle of "ecology in priority, agriculture in second, and construction in postponing". First, all areas with high ecological suitability values were classified as the ecological conservation protection renovation type. Then, the classification was performed according to the agriculture and construction suitability. In this paper, it is divided into six categories, namely: ecological conservation protection renovation type (ECPRT), agricultural production reclamation renovation type (APRRT), construction and development upgrading renovation type (CDURT), ecological agriculture composite renovation type (EACRT), agricultural construction composite renovation type (ACCRT), and balanced development comprehensive renovation type (BDCRT). The dominant types were judged item by item via the idle

residential renovation type identification system, and the renovation types were reasonably classified in conjunction with expert consultation analysis, which provides a theoretical reference frame for the classification promoting the renovation of the idle residential land.

*2.3. Research Methods*

2.3.1. Method for Evaluating the Renovation Potential of Rural Idle Residential Land

According to the relevant regulations of the *Notice on Accelerating the Registration of Confirmation for the Right to Use Residential Land and Collective Construction Land* (No. (2020) 84 issued by the Ministry of Natural Resources) [37] of the Ministry of Natural Resources of China, it is proposed that the registration of confirmation for the right to use land should not be approved for situations such as building houses on illegally occupied cultivated land, building houses in violation of the red line control requirements of ecological protection, the illegal purchase of residential land and small houses by urban residents, etc., and legalizing illegal land use through registration should be prohibited. Combining the data of confirmation for the right to use rural residential land in Yuzhong County, the land use data of rural residential land in the third land survey, remote sensing image data, and on-the-spot investigation, this paper constructed the identification matrix of the idle residential land, which mainly includes seven scenarios (Table 1):

**Table 1.** Identification matrix of the idle residential land.

| Number | Whether the Residential Land Is Confirmed Right | Whether There Are Residential Land Figure in the Third Land Survey Data | Whether There Are Remote Sensing Images of Residential Land | Whether It Is Idle Residential Land |
|---|---|---|---|---|
| 1 | Yes | Yes | Yes | No |
| 2 | No | Yes | Yes | Yes |
| 3 | No | Yes | No | Yes |
| 4 | No | No | Yes | Yes |
| 5 | Yes | Yes | No | No |
| 6 | Yes | No | Yes | No |
| 7 | Yes | No | No | No |

Finally, the rural residential land that meets the following conditions was determined as idle residential land:

(1) The land-use data of the rural residential land has residential land map patches in the third land survey, and the remote sensing image has actual residential land map patches, but the rural residential land has not yet been confirmed;

(2) The land-use data of the rural residential land has residential map patches in the third land survey, and the remote sensing image has no actual residential land map patches, but the rural residential land has not yet been confirmed;

(3) The land-use data of the rural residential land has no residential land map patches in the third land survey, and the remote sensing image has actual residential land map patches, but the rural residential land has not yet been confirmed.

Combined with the research method, we define the concept here: rural idle residential land renovation potential refers to the area of idle residential land available for renovation in rural areas.

Using the symmetrical difference tool in the analysis module of ArcGIS software, the symmetrical difference of the rural residential land data in the third land survey data and the rural residential ownership confirmation data were performed, and the vector data of the possible idle residential land patches were obtained, which were superimposed with the remote sensing image. The vector map patches of the residential land that are not confirmed and exist independently in the third land use survey and the vector map patches that exist in the remote sensing image and can be judged as residential land were determined as the idle residential land. In order to further visualize the spatial agglomeration degree [38,39] of the idle residential land in Yuzhong County, the spatial agglomeration analysis of the

idle residential land was carried out by using the kernel density processing tool of ArcGIS software, and the formula is as shown below:

$$f_n(x) = \frac{1}{nh} + \sum_{i=1}^{n} k\left(\frac{X - X_i}{h}\right). \tag{1}$$

In Formula (1), $f_n(x)$ is the distribution density value of the idle residential land, $(X - X_i)$ is the distance from appraisal point x to event $X_i$, $h$ is the belt width, and $n$ is the number of points in the belt width.

### 2.3.2. Method for Evaluating the Rural Land Suitability

Production, ecology, and life are the three most fundamental functions in the rural district, which correspond to the three major activities, i.e., agricultural production, ecological protection, and construction and development. Therefore, the direction of the idle residential renovation in Yuzhong County mainly focused on three aspects, i.e., ecological protection suitability, agricultural production suitability, and construction and development suitability, for constructing an indicator system (Table 2).

(1)　Evaluation of the ecological protection suitability

**Table 2.** Evaluation indicator system of the rural land suitability.

| Target Layer | Index | Definition | Attribute |
|---|---|---|---|
| Ecological protection suitability | The importance of water and soil conservation function. | It is mainly to prevent and control water and soil loss, protect, improve and make rational use of water and soil resources, and maintain and improve land productivity. | + |
| | The importance of water source conservation function. | It is an important function of ecosystem services, mainly rainwater storage, runoff regulation, and so on. | + |
| | The importance of biodiversity function. | It is the role of ecosystems in maintaining the diversity of genes, species, and ecosystems. | + |
| | The sensitivity of soil and water loss. | It is the sensitivity of the ecosystem to human activities, which reflects the possibility of ecological imbalance and ecological environment problems. | + |
| Agricultural production suitability | The cultivated land quality grade. | The cultivated land grade situation of Yuzhong county. | + |
| | Land use types. | Land use types in Yuzhong County. | + |
| | Rainfall. | Rainfall in Yuzhong County. | + |
| | Distance from the cultivated land. | The distance of the idle residential land from the cultivated land. | − |
| Construction and development suitability | Surface relief. | The difference between the altitude of the highest point and the altitude of the lowest point in the unit. | − |
| | Distance from main roads. | The distance of the idle residential land from the road. | − |
| | Slope. | Degree of steepness and gentleness of land surface unit. | − |
| | Road network density. | The density of road network in Yuzhong County. | + |

The ecological protection suitability (*EPS*) includes two aspects: the evaluation of the importance of ecological system service function and ecological sensitivity [40]. The evaluation was performed by selecting 4 secondary indicators, namely the importance of water and soil conservation function, the importance of water source conservation function, the importance of biodiversity maintenance function, and the sensitivity of soil and water loss, and the calculation formula is as follows:

$$EPS = W_1 S_{pro} + W_2 WR + W_3 S_{bio} + W_4 SS_i \tag{2}$$

In Formula (2): *EPS* is the evaluation value of the ecological protection suitability; $S_{pro}$ is the standardized value of the importance of water and soil conservation function; *WR* is the standardized value of the water source conservation; $S_{bio}$ is the standardized value of biodiversity; $SS_i$ is the standardized value of the sensitivity of soil and water loss; $W_1$–$W_4$ are determined by means of the Delphi method.

The importance of water and soil conservation function was calculated by the following formula [41,42]:

$$S_{pro} = NPP_{mean} \times (1 - K) \times (1 - F_{slo})$$ (3)

In Formula (3): $S_{pro}$ is the service function index of water and soil conservation; $NPP_{mean}$ is the average vegetation net primary productivity for many years, which was obtained by downloading MODIS 250m data from the geospatial data cloud platform; *K* is the soil erodibility factor; $F_{slo}$ is the surface relief.

The water source conservation service function was calculated by the following formula [42,43]:

$$WR = NPP_{mean} \times F_{sic} \times F_{pre} \times (1 - F_{slo})$$ (4)

In Formula (4): *WR* is the index of the ecosystem water source conservation service capability; $F_{sic}$ is the soil seepage factor. According to the USDA soil texture classification, 4 soil texture types of Yuzhong County were each assigned equally between 0 and 1, and the grid map of the soil seepage factor in Yuzhong County was obtained by Kriging interpolation; $F_{pre}$ is the average annual precipitation, which was obtained by interpolating the precipitation data of the meteorological stations around Yuzhong County.

The biodiversity was calculated by the following formula [42]:

$$S_{bio} = NPP_{mean} \times F_{pre} \times F_{tem} \times (1 - F_{alt})$$ (5)

In Formula (5): $S_{bio}$ is the index of the biodiversity service capacity; $F_{tem}$ is the average annual temperature, which was obtained by interpolating the temperature data of the meteorological stations around Yuzhong County; $F_{alt}$ is the altitude factor.

According to the basic principle of the general water and soil loss equation, the indicators such as rainfall erosivity, soil erodibility, slope length, and land vegetation cover are selected [44,45]. The sensitivity of soil and water loss was calculated according to the basic principle of the general soil and water loss equation [42]. The calculation formula is as follows:

$$SS_i = \sqrt[4]{R_i \times K_i \times LS_i \times C_i}$$ (6)

In Formula (6): $SS_i$ represents the index of the sensitivity of soil and water loss; $R_i$ is the rainfall erosivity; $K_i$ is the soil erodibility; $LS_i$ represents the slope length and slope gradient; and $C_i$ represents the vegetation coverage.

(2)   The evaluation of the agricultural production suitability

The agricultural production suitability (*APS*) mainly included 4 secondary indicators: the evaluation of cultivated land quality grade, the evaluation of land-use types, the evaluation of distance from surrounding areas to the cultivated land, and the evaluation of climate conditions, and the calculation formula is as follows:

$$APS = W_5 CLQG + W_6 LUT + W_7 YAR + W_8 DFCL$$ (7)

In Formula (7): *APS* is the evaluation value of the agricultural production suitability; *CLQG* is the standardized value of the cultivated land quality grade; *LUT* is the standardized value of the land use types; *YAR* is the standardized value of the average annual rainfall; *DFCL* is the standardized value of the distance from the surrounding areas to the cultivated land; $W_5$–$W_8$ are determined by means of the Delphi method. *CLQG* could be classified into 12 grades according to the obtained survey data of land foundation status; *LUT* was classified into three categories: agricultural land, construction land, and unused land according to the criterion of the *Classification of Land Use Status* (GB/T21010-2017).

*DFCL* was obtained by using the cost–distance module in ArcGIS to obtain the distance from different segments of residential land in the whole district to the cultivated land.

(3)    The evaluation of the construction and development suitability

The construction and development suitability (*CDS*) are mainly affected by topography conditions and the accessibility of surrounding roads. The evaluation was performed by selecting 4 secondary indicators, i.e., surface relief, distance from main roads, slope gradient, and road network density, and the calculation formula is as follows:

$$CDS = W_9 SU + W_{10} DR + W_{11} SL + W_{12} RD \tag{8}$$

In Formula (8): *CDS* is the evaluation of the construction and development suitability; *SU* is the standardized value of surface relief; *DR* is the standardized value of the distance from main roads; *SL* is the standardized value of slope gradient; *RD* is the standardized value of the road network density; $W_9 \sim W_{12}$ were determined by means of the Delphi method. *SU* and *SL* are extracted by DEM; *DR* was obtained by using the cost–distance module in ArcGIS to get the distance from different places in the whole district to the roads; *RD* was obtained by using the line density analysis tool in ArcGIS.

### 2.3.3. Method for Classifying the Idle Residential Land Renovation Types

From the perspective of intensive and economical use of land, the demand and direction of rural idle residential renovation in different regions were analyzed based on the evaluation of the ecological protection suitability, agricultural production suitability, and construction and development suitability in the county, providing land and other elements for agricultural production, rural construction, and ecological protection. The suitability of rural idle residential renovation is the result of reflection by multiple sub-indicators. It is difficult to directly distinguish the high (H), medium (M), and low (L) value areas of suitability, and there is no unified or universally accepted classification standard. To more reasonably classify the idle residential renovation types, the ecological protection suitability, agricultural production suitability, and construction and development suitability in Yuzhong County were classified into three grades: high, medium, and low, by using the natural discontinuity point method. Considering the practicability of the final results, the rural residential renovation types were determined according to the results of suitability grade classification by reference to the method of identifying dominant types proposed by relevant scholars [46] and in conjunction with the natural geography, socioeconomic and administrative boundaries of Yuzhong County [22] (Figure 3).

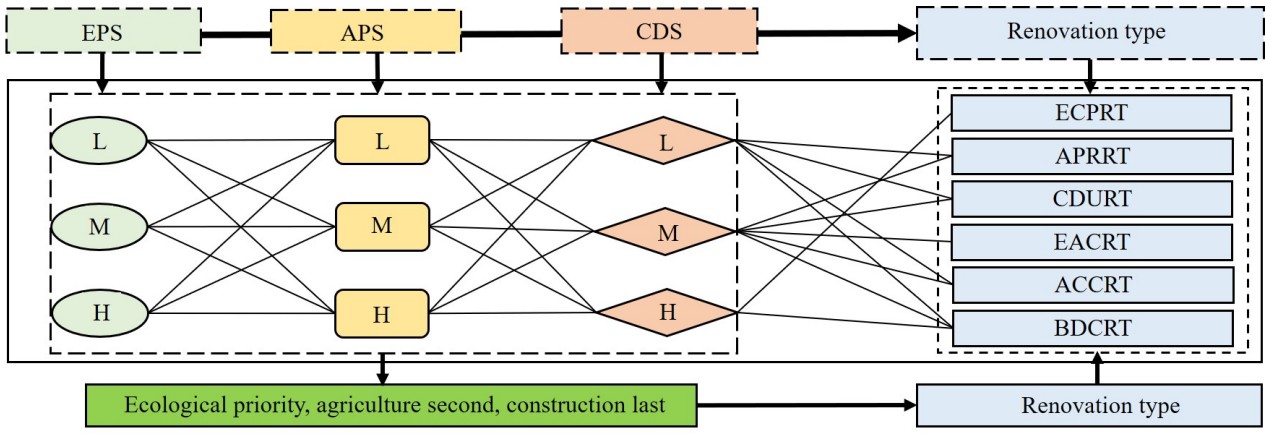

**Figure 3.** Classification of the idle residential renovation types.

(1) The ecological conservation protection renovation type: This renovation type mainly includes H-L-M, H-L-L, H-H-L, H-L-H, H-M-L, H-M-M, H-M-H, M-L-L, H-H-M residential renovation. This means the ecological protection suitability of the area of

the idle residential land is high and its ecological value is high, and therefore, the idle residential land should be renovated as ecological land and intensively protected from ecological damage.

(2) The agricultural production reclamation renovation type: This renovation type mainly includes L-H-L, M-H-L, L-H-M, M-H-M, -M-L residential renovation. This means that areas with agricultural production suitability in the area of the idle residential land is high, the agricultural planting conditions are good, and the idle residential land should be reclaimed as agricultural production land.

(3) The construction and development upgrading renovation type: This renovation type mainly includes L-L-M, L-L-H, L-M-H, M-L-H, M-M-H, L—M, M-L-M residential renovation. This means the construction and development suitability of the area of idle residential land is relatively high, human activities are convenient and the area is suitable for construction and development, and hence the idle residential land should be used for construction and development in order to improve the benefits of land use.

(4) The ecological agriculture composite renovation type: This renovation type mainly includes M-M-L residential renovation. This means that both the ecological and agricultural suitability of the area of idle residential land is high, and it is suitable to renovate the idle residential land into a land-use type for ecological and agricultural composite use.

(5) The agricultural construction composite renovation type. This renovation type mainly includes M-H-H, L-H-H, L-M-M residential renovation. This means that both the agriculture and construction suitability in the area of idle residential land is high, and it is suitable to renovate the idle residential land into the land-use type for agriculture and construction.

(6) Balanced development comprehensive renovation type. This renovation type mainly includes L-L-L, M-M-M, H-H-H, L—L residential renovation. This means that the suitability value of ecology, agriculture, and construction in the area of idle residential land is at the same level. Integrated and comprehensive measures should be taken to the suitable renovation of the idle residential land into a land-use type for comprehensive ecology, agriculture, and construction use.

## 3. Results

### *3.1. Current Situation and Renovation Potential of the Rural Residential Land*

#### 3.1.1. Basic Information of the Rural Residential Land

According to the data of the third land survey, there were 27,531 rural residential patches in Yuzhong County in 2019, with a total area of 56.37 km$^2$, accounting for only 1.71% of the total area of the county (Figure 4a). Among them, Xinying and Zhonglianchuan had the largest number of rural residential patches, with 2089 and 2043, respectively, while Qingcheng, Jinya, and Haxian had relatively few rural residential patches, with 399, 701, and 748, respectively. In the same year, a total of 95,938 residential lands was confirmed in the rural residential ownership confirmation data, with a total area of 30.60 km$^2$, accounting for only 0.93% of the total area of the county, with an average area of the residential land of 318.96 m$^2$ (Figure 4b). Among them, Xiaguanying and Jinya had the largest amount of confirmed residential land, with 9137 and 8340, respectively, while Haxian, Weiying, and Shanghuacha had relatively few confirmed residential lands, with 1291, 1580, and 1938, respectively. By comparison, it was found that the unconfirmed residential land reached 25.77 km$^2$, accounting for 45.72% of the residential land in the third land survey, indicating that there is an obvious phenomenon of "One Household with Two Houses" or "One Household with Multiple Houses" in Yuzhong County.

#### 3.1.2. Identification Results of the Idle Residential Land

Based on the constructed identification method of the idle residential land, it was identified that there are 1700 units of rural residential land to be renovated in Yuzhong County, among which Zhonglianchuan, Weiying, Xinying, Gongjing, Longquan, and Chengguan have the largest number of rural residential land units to be renovated, with

161, 145, 120, 117, 115, and 110 respectively, while Gaoya, Yuanzicha, and Jinya have the least number of idle residential land units, with 42, 48, and 48, respectively (Figure 5a). In space, the idle residential land is mainly decreasing around Chengguan, Zhonglianchuan, Weiying, Longquan, and Shanghuacha as the core, showing a "V-shaped" spatial pattern. The primary distribution area of idle residential land is the southern part of the county along Chengguan, Xiaokangying, Xinying, and Longquan from northwest to southeast, and the secondary distribution area of idle residential land is the Shanghuacha, Zhonglianchuan, and Weiying extending from north to south (Figure 5b).

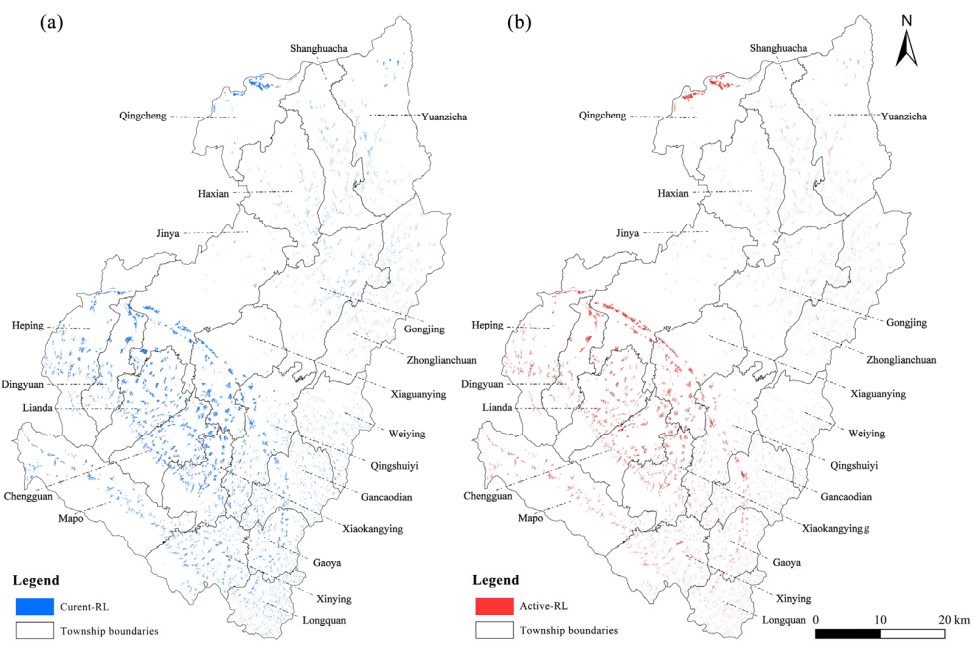

**Figure 4.** The current situation of the rural residential land in Yuzhong County: (**a**) the distribution of the residential land in the third land use survey data; (**b**) the distribution of the confirmed rural residential land.

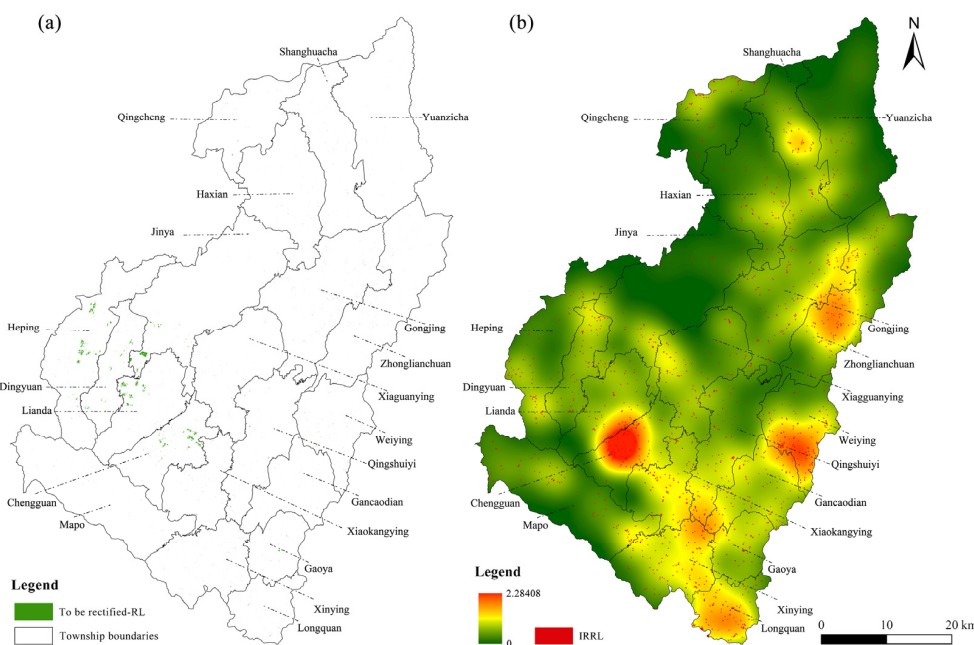

**Figure 5.** The spatial distribution of the idle residential land in Yuzhong County: (**a**) the spatial distribution of the idle residential land; (**b**) the kernel density analysis of the idle residential land.

*3.2. The Evaluation of the Rural Land Suitability*

3.2.1. Evaluation of the Ecological Protection Suitability

The high-value areas of the ecological protection suitability are mainly distributed in the Xinglong Mountain area in the south, and the low-value areas are mainly distributed in Yuanzicha, Shanghuacha, Zhonglianchuan, Gongjing, and Longquan in the north and southeast (Figure 6a). It can be seen that in the high-value areas of the importance of water and soil conservation, the importance of water source conservation and biodiversity are basically distributed in the north of Mapo and the south of Dingyuan, Heping, Lianta, and Chengguan, and the farther away from this center, the lower the value; the level in the eastern and northern parts of the county is the lowest (Figure 6d). The high-value areas of the sensitivity of soil and water loss are mainly distributed in the north of Jinya and Xiaguanying, as well as Mapo, while the low-value areas are mainly distributed in the south of Jinya and Xiaguanying and the north of Heping, Dingyuan, Lianta, Chengguan, and Xiaokangying (Figure 6e).

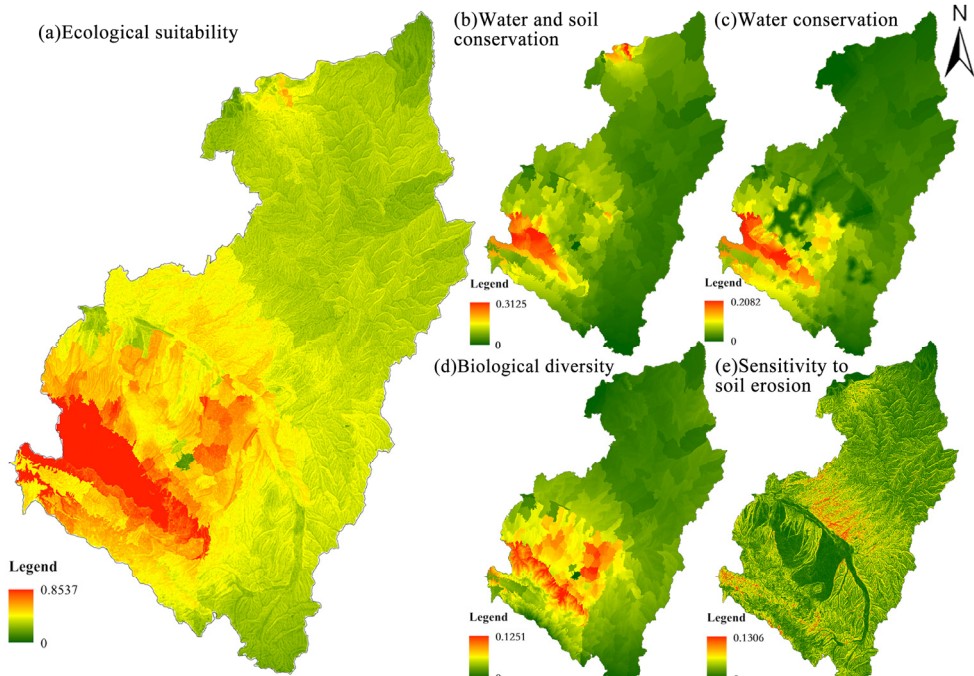

**Figure 6.** The spatial distribution of the ecological protection suitability.

3.2.2. Evaluation of the Agricultural Production Suitability

The high-value areas of the agriculture production suitability are mainly distributed in most areas of the southern and central regions, and the low-value areas are mainly distributed in Jinya, Haxian, Qingcheng, and Yuanzicha (Figure 7a). The results of the agricultural production suitability showed that the quality grade of cultivated land is higher in the central region but is worse in the mountainous and hilly regions in the north and south. The high-value areas are mainly distributed in Qingcheng, Lianta, Chengguan, Qingshuiyi, Jinya, Xiaguanying, Dingyuan, Heping, and Xiaokangying, while the low-value areas are mainly distributed in the southern mountainous areas and the hilly and gully regions in the central region (Figure 7b). Regarding the land use type, the high-value areas are dominant in general, and the low-value areas only appear in the central region. The central region is flat, which is the main urban construction area and an important agricultural production area. The high-value areas are mainly distributed in Chengguan, Xiaokangying, Xiaguanying, and other towns, while the low-value areas are distributed in the towns Chengguan, Xiaokangying, Xiaguanying, etc. (Figure 7c). The average annual precipitation decreases from southeast to northwest successively, with more precipitation in the southern forest region and less precipitation in the northern loess hilly region (Figure 7d).

Regarding the distance from cultivated land, the low-value areas are dominant and are widely distributed; the high-value areas appear only in a few regions and are mainly distributed in the north and south of Xinglong Mountain, the southwest of Qingcheng Mountain, the middle of Jinya, and the north of Heping (Figure 7e).

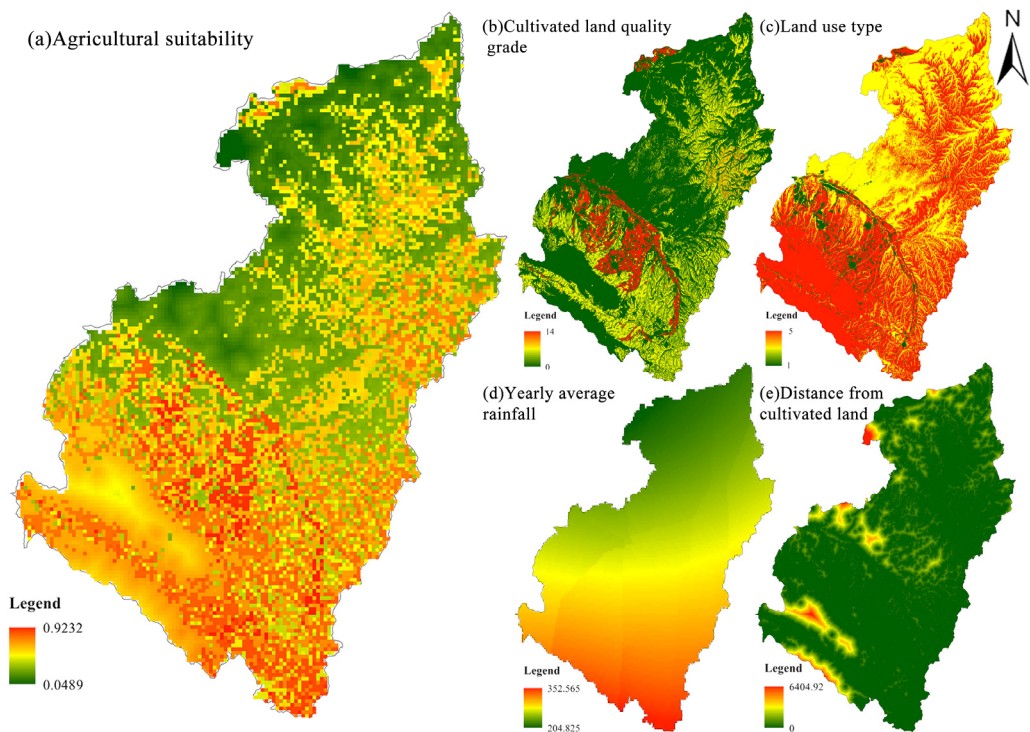

**Figure 7.** The spatial distribution of the agricultural production suitability.

### 3.2.3. Evaluation of the Construction and Development Suitability

The high-value areas of the construction and development suitability are mainly distributed in most areas of Qingcheng, Haxian, Shanghuacha, Jinya, Gaoya, Gancaodian, Chengguan, and Lianta, and the low-value areas are mainly distributed in Mapo and Xinglong Mountain areas (Figure 8a). The results of the construction and development suitability showed that the high-value areas of the surface relief are mainly distributed in the Xinglong Mountain forest area in the south and the Loess hilly arid mountainous area in the north, while the low-value areas are mainly distributed in Heping, Dingyuan, Lianta, Chengguan, Jinya, Xiaguanying, and Xiaokangying (Figure 8b). With the distribution of the road network, the high- and low-value areas away from roads are distributed in scattered grids in various towns and villages, among which the high-value areas are dense in Heping, Dingyuan, Lianta, Chengguan, Xiaguanying, Xiaokangying, Mapo, and Xinying (Figure 8c). The distribution range of the high- and low-value areas of the slope gradient and the high- and low-value areas of the surface relief is roughly the same (Figure 8d). The high-value areas of the road density are mainly distributed in Qingcheng, Haxian, Shanghuacha, Jinya, Heping, Gancaodian, Gaoya, Gongjing, and Zhonglianchuan (Figure 8e).

### 3.3. Classification of Rural Residential Land Renovation Types and Potential Evaluation of Each Type

According to the method for the classification of the idle residential renovation types, the idle residential renovation types in Yuzhong County were determined by summarizing the identification of the idle residential land and the results of the suitability evaluation of rural land (Figure 9).

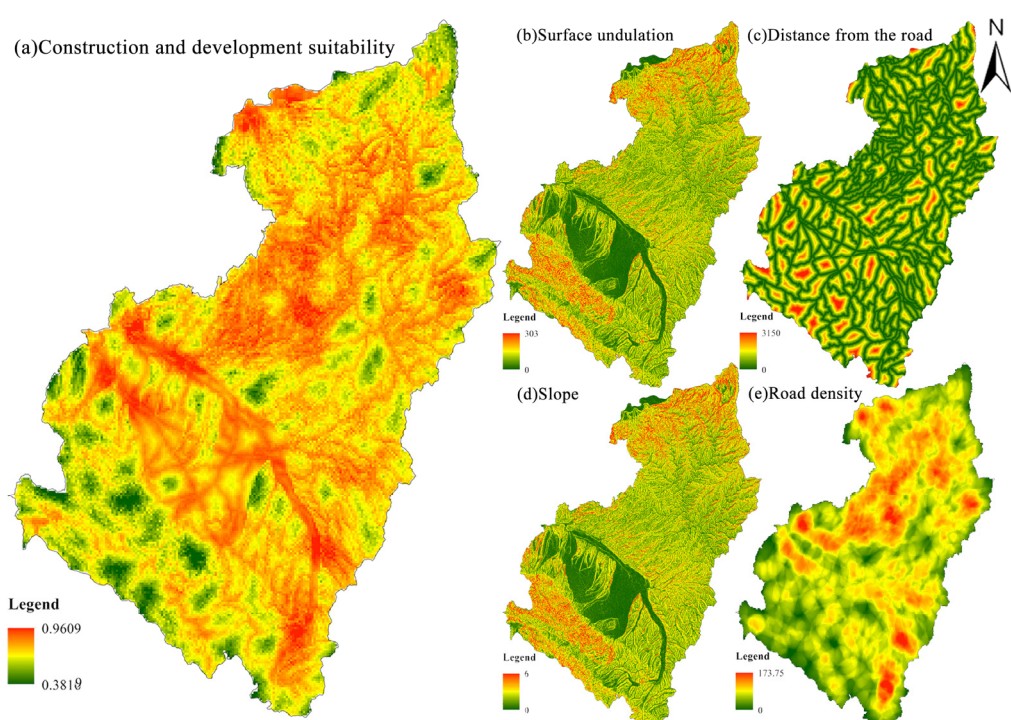

**Figure 8.** The spatial distribution of the construction and development suitability.

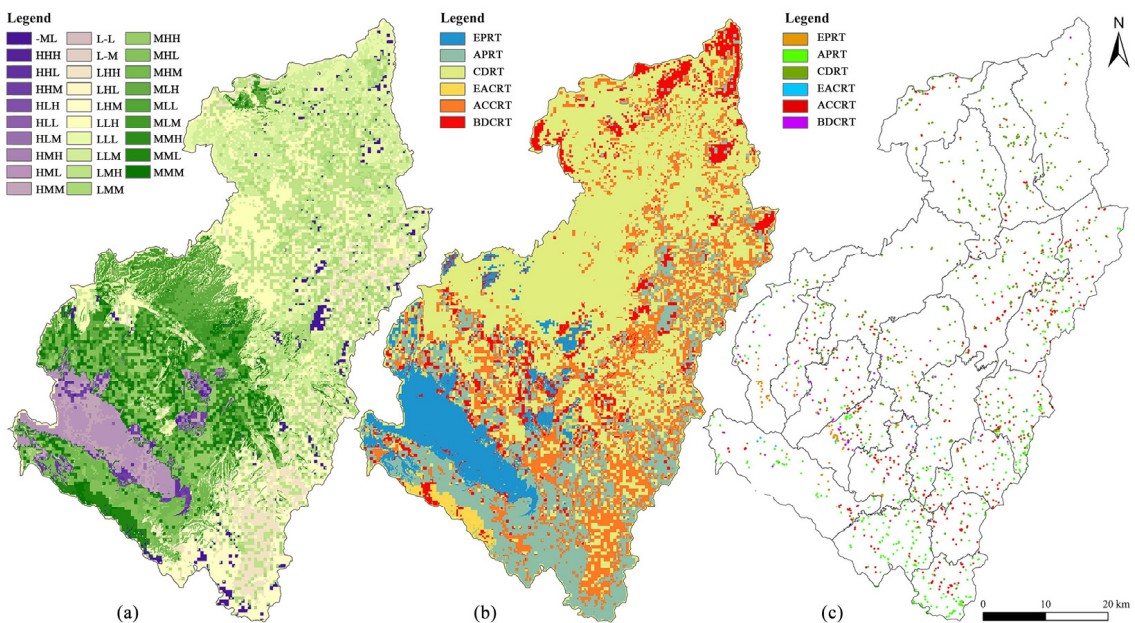

**Figure 9.** The spatial analysis of the idle residential renovation types: (**a**) the spatial coupling analysis of suitability; (**b**) the spatial analysis of renovation types; (**c**) the spatial distribution diagram of the idle residential renovation types.

Among the six renovation types, the construction and development upgrading renovation type have the largest number of idle residential land units, reaching 645, followed by the agriculture production reclamation renovation type and the agricultural construction composite renovation type, which have 442 and 437 idle residential land units, respectively. There are 93 idle residential land units for the ecological conservation protection renovation type, 71 idle residential land units for the balanced development comprehensive renovation type, and only 12 idle residential land units for the ecological agriculture composite reno-

vation type, which is the least in number (Table 3). Among them, the LMH and LLH types are dominant for the construction and development upgrading renovation type, with 291 and 137 idle residential land units, respectively, which are mainly distributed in the central river valley and the northern hilly and mountainous areas such as Qingcheng, Haxian, Shanghuacha, and Yuanzicha (Table 4). The LHM and MHM types are dominant for the agricultural production reclamation renovation type, with 219 and 115 idle residential land units, respectively, which are mainly distributed in towns such as Mapo, Xinying, Longquan, Qingshuiyi, and Xiaokangying. The LHH and LMM types are dominant for the agricultural construction composite renovation type, with 168 and 160 idle residential land units, respectively, which are mainly distributed in the most areas in the east and central regions such as Lianta, Xiaokangying, Gancaodian, Gaoya, Longquan, Qingshuiyi, Gongjing, and Zhonglianchuan. The HMM and HHM types are dominant for the ecological conservation protection renovation type, with 25 and 26 units of idle residential land, respectively, which are mainly distributed in Xinglong Mountain and the north and south regions, and the southern regions of Heping, Dingyuan, Lianta, Chengguan, and Xiaokangying, and the northern regions of Mapo. The MMM type is dominant for the balanced development comprehensive renovation type, reaching 56 idle residential land units, which is mainly distributed in Qingcheng, Yuanzicha, Shanghuacha, and other towns; the MML type is dominant for the ecological agriculture composite renovation type, with 12 idle residential land units, which is mainly distributed in the south of Mapo.

**Table 3.** Statistical table of the idle residential renovation types.

| Type | Area (m²) | Subclass | Number of Residential Land |
|---|---|---|---|
| Construction and development upgrading renovation type (645) | 439,600 | LLM | 71 |
| | | LLH | 137 |
| | | LMH | 291 |
| | | MLH | 36 |
| | | MMH | 78 |
| | | L-M | 2 |
| | | MLM | 30 |
| Ecological agriculture composite renovation type (12) | 5400 | MML | 12 |
| Agricultural construction composite renovation type (437) | 333,600 | MHH | 109 |
| | | LHH | 168 |
| | | LMM | 160 |
| Balanced development comprehensive renovation type (71) | 44,800 | LLL | 7 |
| | | MMM | 56 |
| | | HHH | 7 |
| | | L-L | 1 |
| Ecological conservation protection renovation type (93) | 73,800 | HLM | 6 |
| | | HLL | 2 |
| | | HHL | 11 |
| | | HLH | 8 |
| | | HML | 4 |
| | | HMM | 25 |
| | | HMH | 7 |
| | | MLL | 4 |
| | | HHM | 26 |
| Agricultural production reclamation renovation type (442) | 278,000 | LHL | 55 |
| | | MHL | 30 |
| | | LHM | 219 |
| | | MHM | 115 |
| | | -ML | 23 |

**Table 4.** Statistical table of the idle residential renovation types in individual towns.

| Town Name | ECPRT | APRRT | CDURT | EACRT | ACCRT | BDCRT |
|---|---|---|---|---|---|---|
| Chengguan | 29 | 25 | 26 | 1 | 17 | 12 |
| Xiaguanying | 6 | 5 | 57 | 0 | 10 | 4 |
| Dingyuan | 9 | 7 | 23 | 0 | 11 | 7 |
| Lianda | 1 | 4 | 24 | 0 | 20 | 8 |
| Heping | 27 | 11 | 21 | 2 | 6 | 8 |
| Gaoya | 0 | 9 | 9 | 0 | 24 | 0 |
| Jinya | 0 | 1 | 34 | 0 | 11 | 2 |
| Qingcheng | 0 | 0 | 37 | 0 | 13 | 0 |
| Gongjing | 0 | 19 | 58 | 0 | 40 | 0 |
| Gancaodian | 0 | 26 | 27 | 0 | 36 | 5 |
| Xinying | 0 | 84 | 11 | 0 | 23 | 2 |
| Qingshuiyi | 2 | 24 | 27 | 3 | 16 | 3 |
| Xiaokangying | 11 | 14 | 7 | 2 | 28 | 12 |
| Longquan | 0 | 77 | 7 | 0 | 31 | 0 |
| Weiying | 1 | 58 | 45 | 1 | 40 | 0 |
| Haxian | 0 | 0 | 51 | 0 | 7 | 0 |
| Mapo | 7 | 42 | 4 | 3 | 23 | 4 |
| Shanghuacha | 0 | 0 | 75 | 0 | 14 | 0 |
| Yuanzicha | 0 | 0 | 45 | 0 | 2 | 1 |
| Zhonglianchuan | 0 | 36 | 57 | 0 | 65 | 3 |
| Total | 93 | 442 | 645 | 12 | 437 | 71 |

According to the identification results of the idle residential land, the potential for future renovation of the residential land of "One Household with Two Houses" and "One Household with Multiple Houses" in Yuzhong County is 1.18 km$^2$. Among them, the construction and development upgrading renovation type are 439,600 m$^2$, the agriculture production reclamation renovation type is 278,000 m$^2$, the agricultural construction renovation type is 333,600 m$^2$, the ecological agriculture composite renovation type is 5400 m$^2$, the ecology conservation protection renovation type is 73,800 m$^2$, and the balanced development comprehensive renovation type is 44,800 m$^2$.

## 4. Discussion

### 4.1. Innovation in Method for the Identification of Idle Residential Land

Rural residential land is the foundation of farmers' life, and the rural residential land expansion in China has been effectively curbed. However, how to revitalize idle residential land has become the key issue of rural revitalization and the key to village renovation. There are general phenomena in rural areas of China such as "One Household with Multiple Houses", "Building New Houses without Demolishing Old Ones", long-term neglect, etc. The continuous idleness of the rural residential land hinders the rural economy, affects social stability and in turn leads to rural decline, which is an important problem facing the coordinated development of urban and rural areas in China in the new period [39]. As the most direct potential source of rural land use, idle residential land is an important reserve land asset for supporting rural reconstruction. However, there is no unified understanding of the definition and identification methods of the rural "idle residential land". Therefore, how accurately identifying idle residential land has become the primary problem of residential renovation [17,19,20]. In terms of the identification method, qualitative analysis or statistical analysis is mostly used as a single method, while comprehensive use of remote sensing, GIS, spatial mathematical models, and other integrated methods is relatively few. Based on the rural residential ownership confirmation data, land use survey data, high-resolution remote sensing images, and on-the-spot investigation, this paper constructed the identification matrix of the idle residential land, which is a new exploration for the identification of the idle residential land and is helpful to improve identification accuracy.

### 4.2. Suitability Is the Foundation of the Idle Residential Land Renovation

Scientific classification of the idle residential renovation types [47] is the premise and foundation for optimizing rural land use structures. Suitability evaluation is a process of judging whether a certain renovation mode or type is suitable and how suitable it is. It is impossible to fundamentally solve the problem of rural land resources without classifying them according to their suitability. There are many factors that affect the layout of rural residential sites. It is necessary to combine the characteristics and conditions of Yuzhong County itself to determine the direction of renovation from three aspects: ecological protection suitability, agricultural production suitability, and construction and development suitability. The purpose of the evaluation of different suitability and the classification of renovation types in the county district is to scientifically identify the functional orientation of land in different districts and to provide the scientific basis for affording full play to the comparative advantages of land use and maximizing the advantages of regional development. The development and utilization of regional land and the allocation of public resources should be based on regional suitability evaluation, pay attention to affording play to comparative advantages, and constantly strengthen specific functions to revitalize rural idle land. The renovation of the idle residential land in Yuzhong County takes full account of the environmental characteristics of the residential land. According to the suitability of the residential land, the idle residential land is classified into different renovation types, and then the classification system of the idle residential renovation types is established, which has important practical significance for guiding village layout planning, coordinating regional development, and ecological protection, and promoting rural revitalization strategy.

### 4.3. The Idle Residential Renovation Boosts Rural Revitalization

The core intent of rural revitalization is to reshape the social economy and spatial structure of rural areas. The renovation of rural idle residential land is the only way to improve the rural living environment and alleviate the urgent need for rural revitalization. First, the rural residential land is related to the fundamental interests of every farmer, and the flow and reorganization of other rural elements. The renovation of rural residential land with multiple houses, new construction but not demolished, and dilapidated and collapsed houses can stimulate the vitality of village development and promote rural revitalization and transformation development. Second, it is of positive significance for the coordinated evolution and sustainable development of the human-land system to study the land response mechanism and obstacle factors of ecological protection, agricultural production, construction, and development in different types of areas. This has reference value for the implementation of the rural revitalization strategy in the new era. Finally, the reform of the efficient rural land use and use system has also become the internal requirement of rural revitalization and urban–rural integrated development. It is the key to responding to the call of national policies, making every effort to realize rural revitalization, and promoting long-term development and construction of rural areas. Residential renovation is helpful in developing industries, expanding the collective economy, and improving the appearance of villages, thereby enhancing the development potential of the countryside and improving the happiness index of villagers.

### 4.4. The Enlightenment of the Idle Residential Renovation on Policy

With the increasing idleness of the rural residential land, we should strengthen the policy control of the rural residential land. The current residential land management exists many drawbacks, such as the management system lacking an exit mechanism, specific idle residential land recovery procedures, a compensation mechanism, etc. [48]. Through the identification of the idle residential land in Yuzhong County and the classification of renovation types, it is found that the key to avoiding a large number of idle residential land units is to establish a perfect residential land approval process from the source, formulate a complete residential land management system, and improve the residential



land exit guarantee mechanism. Therefore, the county government departments should standardize the examination and approval procedure of rural residential land for house-building applications [49] according to the *Notice on Regulating the Examination and Approval Management of Rural Residential Land* jointly issued by the Ministry of Agriculture and Rural Affairs and the Ministry of Natural Resources. At the same time, the county government needs to allocate the county's construction land index as a whole according to the county population's mobility and formulate county residential land management measures in conjunction with residential land use and the idle situation in the county and the population activities in the county [50]. For the residential land that is determined to be necessarily withdrawn or renovated, we should formulate differentiated withdrawal policies according to the spatial heterogeneity of the idle residential land, the family situation of farmers, the willingness to withdraw, and the preference for property rights of houses, etc., so as to maintain the internal stability of rural areas [47]. Instead of forcibly and blindly implementing urban–rural construction land increase/decrease linkage projects, we should take the spatial planning of the country as a guide, carry out a good job in village planning and land improvement planning of the whole area, formulate scientific plans for the remediation and reuse of rural "hollow houses", gradually adjust and optimize the layout of rural settlements by farming as appropriate, greening as appropriate, and constructing as appropriate to improve the efficiency of comprehensive use of rural land, stimulate the vitality of rural land resources, and promote the revitalization of the countryside and the improvement of the rural living environment.

## 5. Conclusions

The land issue is the key to the implementation of the rural revitalization strategy. Rural residential renovation is the key to land renovation planning, and the weak link and implementation difficulty of land renovation. The identification standard and calculation method of idle residential land are the premise for determining the idle degree of the residential land. In this paper, the identification matrix of idle residential land was constructed by integrating multi-source data, such as rural residential ownership confirmation data, land use survey data, high-resolution remote sensing images, and field research. The results show that there are 1700 rural residential land units to be renovated in Yuzhong County. There are significant phenomena of "one household with two houses" and "one household with multiple houses" in Yuzhong County. In addition, the suitability of ecological protection, agricultural production, and construction and development directly affects the planning layout and renovation direction of rural residentials. The coordination of the spatial relationship between the rural idle residential land and the suitability of ecological protection, agricultural production, and construction and development determines the spatial basis for the classification of internal transformation and renovation types of the rural idle residential land. Based on the suitability evaluation, we classified the rural idle residential land into six types: ecological conservation and protection type, agricultural production reclamation type, construction and development upgrading type, ecological agriculture composite type, agricultural construction composite type, and balanced development comprehensive type. In the future, the government should do a good job in doing a solid job in rural residential renovation planning, formulate a scientific plan for the renovation, relocation, and reuse of rural "hollow houses". Adapting to local conditions, the government should gradually adjust and optimize the layout of rural residential land, improve the comprehensive utilization efficiency of rural land, stimulate the vitality of rural land resources, and promote the rural revitalization of rural areas and the improvement of rural living environment.

There are still some shortcomings in this paper: First, due to the limitation of data sources, the data studied in this paper only use the rural residential ownership confirmation data and land use survey data in a single year. Most idle houses are not idle all year round, and seasonal idleness is common. Therefore, only the "illegal building" of "One Household with Two Houses" and "One Household with Multiple Houses" is considered, which has a

certain impact on the analysis of spatial distribution characteristics of the "idle residential land". Second, this paper puts forward that the direction of idle residential land renovation is determined based on the suitability of rural land, and the subjective factors such as government behavior and individual/family willingness are not analyzed sufficiently. Idle residential land renovation is implemented under the impetus of the government. The impact analysis of government-supporting policies and the analysis of villagers' satisfaction after implementation have important guiding significance for formulating the policy of idle residential land renovation, which is worthy of further discussion in the future. Finally, we need to further consider the local practice, summarize the activation mode of rural idle residential land, and construct the path and guarantee mechanism to promote the efficient use of the rural idle residential land in different situations.

**Author Contributions:** Conceptualization, L.M.; Data curation, T.T.; Formal analysis, L.M.; Investigation, Y.Y. and Y.L.; Methodology, T.T.; Project administration, L.M.; Resources, L.M.; Software, Y.Y. and Y.L.; Validation, Y.Y. and Y.L.; Visualization, T.T.; Writing—original draft, T.T. All authors have read and agreed to the published version of the manuscript.

**Funding:** This research was funded by the National Natural Science Foundation of China (grant number 41961033) and the Science and Technology Major Special Program Projects of Gansu Province (grant number 22ZD6WA057).

**Institutional Review Board Statement:** Not applicable.

**Informed Consent Statement:** Not applicable.

**Data Availability Statement:** No new data were created or analyzed in this study. Data sharing is not applicable to this article.

**Acknowledgments:** Authors express great thank to the financial support from National Natural Science Foundation of China and the Science and Technology Major Special Program Projects of Gansu Province.

**Conflicts of Interest:** The authors declare no conflict of interest.

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
