# Peer review of "Renovation Potential Evaluation and Type Identification of Rural Idle Residential Land: A Case Study of Yuzhong County, Longzhong Loess Hilly Region, China"

_land, doi:10.3390/land12010163_

Round 1

Reviewer 1 Report

The paper Type identification and potential evaluation of rural idle residential renovation: A case study of Yuzhong County, Longzhong Loess Hilly Region, China fits well to the topic and scope of the special issue „Land Consolidation and Rural Revitalization" since the article deals with the evaluation of idle rural land from the aspects of ecological protection suitability, agricultural production suitability, and construction and development suitability in a certain area of China.

The manuscript is clear and relevant to the field hence rural residential renovation is a key element in rural development.

The used methods are appropriate, and the manuscript’s results are reproducible based on the details given in the section methods. The analysed data is adequate.

The conclusions are consistent with the evidence and arguments presented in the previous chapters.

The authors processed the recent literature in the field. 32 out of 65 citations are from the last 5 years. Self-citations are within the limit.

The English language of the manuscript is appropriate, the paper is easy to follow.

In summary, I recommend the acceptance of the manuscript after minor revision.

Specific comments:

The paper is inconsequent in the use of hm2 (e.g., row 18 and 601) and km2 (e.g., row 465) I recommend using the km2 throughout the paper since it is easier to understand.

Presenting the socio-economic profile of the region authors have used the form 46.11x104 on several occasions which makes reading and understanding the numbers difficult. The size of the numbers does not justify this form.

Also in this section, presenting the economic situation yuan is used but considering the international audience of the journal the use of USD is recommended.

The figures and tables are appropriate and help to understand the main text. However, the abbreviations in Figure 2: ECPRT, APRRT, CDURT, etc. are difficult to follow at the first sight. (Authors use the same abbreviations in the text, too.) Please re-think the use of the above-mentioned abbreviations in both cases.

The paper is not consistent in some cases in the spelling of geographical areas. For example, using the name ZhongLianchuan in Figure 4. but using the spelling Zhonglianchuan in-text (row 485). This applies to all other labels in Figure 4. and Figure 5.

Reviewer 2 Report

My comments on the manuscript are as follows:

1. (Line 2)From the perspective of the title, the manuscript includes two parts: type identification and potential evaluation. However, the manuscript contains two core contents: identifying the rural idle residential land and classifying the idle residential land renovation types, which are seriously inconsistent with the title of the manuscript.

2. (Line 30)In the introduction section, the literature review part only made a general introduction to the research on idle residential land but did not make a specific elaboration on the potential evaluation and type identification. Therefore, the innovation of the manuscript cannot be supported. In addition, some of the sentences in the literature review part have poor logic (Lines 100 and 104~105), which further weakens its support for the manuscript.

3. (Line 130)Much of the content in this part is redundant and has little relevance to the topic of the manuscript. In addition, some language expressions need to be improved. For example, the expression "Research Area Profiles" is rarely seen in papers.

4. (Line 177)The data source section lacks an introduction to data accuracy, resolution, and time, especially the remote sensing image data, which is one of the key data in the article. Information such as resolution, classification method, and classification system of remote sensing images should be introduced. In addition, a further introduction should be made to the rural residential ownership confirmation data and the third land survey data, especially how to deal with the problems of accuracy and land use classification between different data sources.

5.(Line 257)The names of the three sections in Figure 5 are inconsistent with the titles of the method and result sections. In addition, remote sensing image data, one of the key data used in the manuscript, is missing in Figure 5.

6. (Line 273)The identification results of idle residential land in Table 1 are completely consistent with the confirmation data. Then, please explain what is the significance of using the third land survey data and remote sensing data.

7. (Line 299)This section is too long, and please compress it to highlight the topic of the article.

8. (Line 408)Please explain how to comprehensively renovate and utilize the idle residential land of the last three types, especially the balanced development and comprehensive renovation type.

9. (Line 426)First, all abbreviations in Figure 3 are not explained in the text. Secondly, Figure 3 cannot effectively explain the identification process of the six types of idle residential land renovation. Finally, please explain the relationship between the word 'remediation' in the figure and the word 'renovation' in the text.

10.(Line 460)This section is titled "Analysis of the current situation of the rural residential land", but this section includes a subsection "Identification results of the idle residential land", which is one of the important innovations described in the manuscript. Is this appropriate?

11.(Line 477)What is the function of this sentence “a land-use type for ecology, agriculture, and construction comprehensive use”?

12.(Line 482)Please explain how to obtain 1,700 idle residential land patches based on the 27,531 rural residential land patches from the third land survey data and the 95,938 residential land patches from the confirmation data. These three numbers of rural residential land patches confuse the readers.

13.(Line 498) The single-factor evaluation has little to do with the topic of the manuscript but is very long, while the comprehensive evaluation is closely related to the topic of the manuscript but is very short.

14.(Line 612)Please explain the difference between the words 'town' and 'township'.

15.(Line 613)First of all, the Discussion section does not provide in-depth analysis and elaboration based on the research results and is almost completely out of touch with the results. Secondly, Sections 4.1 and 4.3 are more like research significance and literature review, which are more suitable for the introduction section. Finally, part of 4.2 is an introduction to some basic concepts, which is more suitable for the method part.

16. Finally, there are a lot of long sentences and some wrong terminology (such as kernel density analysis) in the manuscript, so the language of the manuscript needs to be further improved.

In general, although the manuscript has certain practical significance, there are still many deficiencies in innovation, research ideas, manuscript structure, and language expression. Therefore, it is recommended that the article be reviewed after a major revision.

Round 2

Reviewer 2 Report

My comments are as follows:

1. The title of the manuscript is inconsistent with the content of the manuscript. The reasons are as follows:

(1) Inconsistencies can be found in the title of the manuscript and the titles of the chapters:

   Title: Type identification and potential evaluation of rural idle residential renovation

   Methods:

           2.3.1. Method for identifying the rural idle residential land

           2.3.2. Method for evaluating the rural land suitability

           2.3.3. Method for classifying the idle residential land renovation types

   Results:

           3.1. Analysis of the current situation of the rural residential land

           3.2. The evaluation of the rural land suitability

           3.3. The classification of the rural residential renovation types and potential evaluation

What is the relationship between "Type identification" in the title, "identifying the rural idle residential land" in the methods, and "classification of the rural residential renovation types" in the results? The confusion about these concepts reflects the author's lack of rigorous research attitude.

(2) A serious problem in the manuscript is that the author confused the relationship between "type identification" and "potential evaluation". In particular, the word "potential" appears 10 times in the text, but none of them explains the specific meaning of "potential". According to the manuscript and the author's reply, we can infer that the area of each type is the so-called "potential". Where is the process of potential evaluation? Is type identification the potential evaluation? Can the potential be evaluated only based on suitability? Is it of practical significance to evaluate the potential lacking of socioeconomic and planning factors? These problems further reflect the author's unclear differences between related concepts.

2. (Line 54-111) Unfortunately, the author avoided replying to my comments. As qualified researchers, we should know that the literature review is used to support the innovation of articles. However, for the possible innovation "type identification and potential evaluation" of this manuscript, the author did not point out the existing research status and shortcomings.

Furthermore, some views of the literature review in the manuscript are ridiculous(for example, "the corresponding renovation zoning is often implemented in county and city units, ignoring the possible differences within them", and "in terms of research methods, most scholars used a single method of qualitative analysis or statistical analysis but lacked the comprehensive use of remote sensing, GIS, spatial math-110 ematical model, and other integrated methods and spatial decision-making"). As far as I know, there are many related research works on the patch scale and using comprehensive methods.

3. (Line 155) Unfortunately, the author avoided replying to my comment again.

(1) First, the authors still have not introduced any information or processing methods of remote sensing data which is key data in the manuscript. Related to this comment, the author chose to avoid replying to the 5th comment (In addition, remote sensing image data, one of the key data used in the manuscript, is missing in Figure 5).

(2) Second, because I have used the rural residential ownership confirmation data and the third land survey data, I want to know how to deal with the problems of different accuracies and land use classification systems between different data sources. This is also the reason why I raised the 12th comment last time. Unfortunately, the author avoided replying to the comment again.

(3) Finally, the sentence "unified all the data to a resolution of 30*30" (What is the number's unit?)in the manuscript is confusing. Because the accuracy of the two key data (the rural residential ownership confirmation data and the third land survey data) is much higher than 30m in this place, it is unimaginable to reduce the data accuracy here. Furthermore, "unified all the data to a resolution of 30*30" also contradicts the patch numbers of the rural residential ownership confirmation data, the third land survey data, and the identification results of the idle residential land in the following content.

4. (Line 223~238) What is the difference between the three words "remediation", "reclamation", and "renovation" in the classification of the idle residential renovation types? Whether the two words "remediation" and "reclamation" in "agricultural production reclamation remediation type" are repeated? This manuscript mixed similar concepts and did not explain the differences between them, further indicating that the authors did not understand the relevant concepts enough.

5.(Line 355) First, the classification has not reflected "the principle of ecological priority, agriculture second, and construction last". Second, "the balanced development and comprehensive renovation type" is confusing, how about "multiple suitable renovation types"? Because there is no reason for comprehensive renovation due to multiple suitability. It is more practical and scientific to choose one specific and main renovation type based on multiple suitability.

6. (Line 372) It is impossible for readers to get the author's classification results based on the connections in Figure 3. Furthermore, the classification has not reflected “the principle of ecological priority, agriculture second, and construction last”. Because if the principle is adopted, how can there be "the balanced development comprehensive renovation type"?

7. (Line 445) I don't agree with the authors' reply on suitability, because I have never seen a detailed introduction of single factor suitability like this manuscript, which is meaningless for the manuscript. The only purpose that the authors do this I can think of is to fill up the length of the manuscript. Please delete the single-factor evaluation section and detail the comprehensive evaluation section.

8. I don't agree with the authors' explanations of the difference between the words “town” and “township”.

9. I don't think the discussion part of the manuscript has been substantially improved. First, the discussion was not based on the research results. For example, the whole manuscript did not elaborate on what different renovation types can do specifically. Secondly, most of the discussion is far from the manuscript's theme "type identification and potential evaluation", and the manuscript's results cannot support the discussion. These are the problems often encountered by most authors who write a paper for the first time.

My final suggestion is to reject the manuscript. My reasons are as follows:

(1) The manuscript has no sufficient reason to support the innovation of the manuscript.

(2) The data introduction of the manuscript is incomplete, and the introduction of potential evaluation methods is lacking, which leads to the non-repeatability of this manuscript.

(3) The manuscript often mixes related concepts in the manuscript, resulting in a poor theoretical basis of the manuscript.

(4) The authors avoided replying to key comments I care about, leading to readers' further doubt about the scientificity of the manuscript.

(5) The language of the manuscript still needs further improvement. For example, what is "1700 rural residential land" mean? There are many problems like this in the manuscript, and it is difficult to list them all.

In conclusion, it can be seen from the manuscript that the author lacks some basic scientific research quality, and this manuscript is more like the work of a scientific research beginner, far from being publishable. Please consider the publication of this manuscript carefully.
